# Idiopathic Pulmonary Fibrosis: Pathogenesis and the Emerging Role of Long Non-Coding RNAs

**DOI:** 10.3390/ijms21020524

**Published:** 2020-01-14

**Authors:** Marina R. Hadjicharalambous, Mark A. Lindsay

**Affiliations:** Department of Pharmacy and Pharmacology, University of Bath, Claverton Down, Bath, BA2 7AY, UK; M.A.Lindsay@bath.ac.uk

**Keywords:** long non-coding RNAs, lncRNAs, fibrosis, lung, IPF

## Abstract

Idiopathic pulmonary fibrosis (IPF) is a progressive chronic disease characterized by excessing scarring of the lungs leading to irreversible decline in lung function. The aetiology and pathogenesis of the disease are still unclear, although lung fibroblast and epithelial cell activation, as well as the secretion of fibrotic and inflammatory mediators, have been strongly associated with the development and progression of IPF. Significantly, long non-coding RNAs (lncRNAs) are emerging as modulators of multiple biological processes, although their function and mechanism of action in IPF is poorly understood. LncRNAs have been shown to be important regulators of several diseases and their aberrant expression has been linked to the pathophysiology of fibrosis including IPF. This review will provide an overview of this emerging role of lncRNAs in the development of IPF.

## 1. Idiopathic Pulmonary Fibrosis

Fibrosis is a pathophysiological condition that can affect nearly every organ in the human body where irregular and excessive accumulation of scar tissue leads to organ failure and potentially death as seen in the final stages of fibrotic diseases such as pulmonary [1], cardiac [2], nephrotic [3], and hepatic fibrosis [4]. In combination with genetic factors, tissue injuries may provoke the development of fibrosis including exposure to damaging environmental stimuli such as irritants, smoke, radiation, viral, and bacterial infections [5,6].

Idiopathic pulmonary fibrosis (IPF) is a progressive chronic interstitial lung disease (ILD) which is characterized by scar tissue accumulation, and therefore thickening of the normal lung walls, leading to impaired gas exchange and restricted ventilation. IPF is a disease of unknown aetiology, which makes the development of effective drug treatments particularly challenging [5]. Nonetheless, scientists have been intensively researching the molecular and cellular mechanisms of the disease and, although the pathogenesis of IPF is still unclear, several theories regarding the pathophysiology of IPF have been proposed [7].

As is the case with most ILDs, inflammation was initially thought to be the major player in IPF until unresponsiveness to anti-inflammatory medications prompted the re-evaluation of this idiom [8,9]. However, the presence of immune cells in IPF lungs has been a consistent pathological finding and could be important in the development of the disease [10,11,12,13,14]. The histology of fibrotic lungs also indicates irreversible accumulation of scarred tissue characterized by collagen deposition and other alterations to the extracellular matrix (ECM) which dramatically remodels the lung architecture by stiffening the distal airspaces and parenchyma [5]. It has been suggested that lung fibrosis could be provoked by a number of different cell types including epithelial cells, fibroblasts, myofibroblasts, and immune cells [1].

## 2. IPF Symptoms

IPF induces exertional dyspnoea, a feeling of breathlessness, which is one of the most common symptoms experienced. Dyspnoea is often accompanied with a non-productive dry cough which progressively worsens over time until it becomes clear to patients that their daily functionality is compromised. Pulmonary malfunction due to fibrosis can induce other symptoms such as inspiratory crackles, chest discomfort, finger clubbing, weakness, and loss of appetite [15,16,17].

## 3. The Pathogenesis of IPF

Under physiological conditions, fibrogenesis is initiated in response to tissue injury and forms part of the wound repair process involved in the restoration of homeostasis. Wound repair is commonly initiated by epithelial injury, leading to activation of the coagulation and inflammation cascades. This in turn results in the activation, recruitment, and proliferation of fibroblasts that are responsible for the release of ECM components. In the final remodeling stage, the wound area is resolved, and normal tissue structure and structural integrity is restored [18].

During fibrosis associated with IPF, any stage of the wound repair process can be dysregulated (Figure 1), resulting in the irreversible accumulation of scar tissue. These fibrotic areas are characterized by an overproduction of ECM components, predominantly collagen as well as other fibrotic proteins, which dramatically remodels the lung architecture and leads to excessive scarring [19]. The secretion of several pro-fibrotic cytokines and growth factors are thought to be crucial mediators of fibrosis. These drive the migration, proliferation, and activation of mesenchymal cells which ultimately result in the differentiation of fibroblasts into α-smooth muscle actin (α-SMA)-expressing myofibroblasts and the development of fibrotic foci, a dense collection of myofibroblasts and scar tissue [20]. The pleiotropic growth factor TGF-β1 (transforming growth factor β1) is regarded as a key player of fibrosis [21] along with other mediators such as platelet-derived growth factor (PDGF) [22], interleukin 1β (IL-1β) [23], as well as several chemokines [24] and cytokines [25].

At the present time, the drivers of the aberrant wound healing process leading to fibrosis are unknown. However, certain cells types such as alveolar epithelial cells and fibroblasts, as well as a dysregulated coagulation and inflammatory response have been implicated in disease initiation and progression [1,26,27,28].

## 4. Alveolar Epithelial Injury

Prior to the activation of fibroblasts, it is believed that type I AECs could have been subjected to repetitive injury, which causes damage to the delicate epithelium structure. Type I AECs line more than 90% of the alveolar surface and damage promotes the activation and proliferation of the surfactant-producing type II AECs [29]. This results in hyperplasia of the type II AECs in order to cover the exposed alveolar surface, as well as the activation of local coagulation pathways and the initiation of a provisional matrix also known as a wound clot [30].

During the normal healing process, the lung tissue eventually regains its original structure and function as the provisional matrix gradually dissipates. In the case of injury of the type I AECs, the hyperplastic type II AECs are thought to undergo regulated apoptosis and trans-differentiation into type I AECs in order to re-establish a fully functional alveolar epithelium [31].

However, if the epithelial basement membrane remains disturbed following extensive damage, the alveoli can collapse and type II AECs fail to undergo re-epithelisation. An aberrant wound repair response can then be initiated during which the epithelial cells, predominantly type II AECs, are thought to release several pro-fibrotic cytokines, growth factors, and other chemokines at the site of injury [32]. The epithelial injury is then thought to ultimately promote the activation and proliferation of fibroblasts and myofibroblasts and the formation of a stiffened ECM in IPF [33,34].

## 5. Inflammation

One of the initial concepts relating to IPF pathogenesis was that the disease was initiated and driven by chronic inflammation. However, the role of inflammation was questioned when anti-inflammatory and immunosuppressant therapies failed to improve lung function and survival of IPF patients. A study of 330 IPF patients showed no effect on progression-free survival when treated with the immune-regulatory cytokine interferon-gamma-1β (IFN-γ-1β) [35]. Additionally, the INSPIRE randomized double-blinded placebo-controlled trial of IFN-γ-1β did not show any benefit as compared with a placebo and was terminated [36]. The IFIGENIA trial, a double-blind, randomized, placebo-controlled study assessed the efficacy of N-acetylcysteine (NAC) added to prednisone and azathioprine [37], however the survival benefit of IPF patients was questioned. The follow-up PANTHER trial was designed to address some of the issues of the IFIGENIA trial and evaluated the response in IPF patients of this triple therapy, but it was terminated prematurely when it showed significantly increased mortality [38]. The study continued as a two-group study (NAC vs. placebo) but demonstrated no benefit to IPF patients [39].

Although the role of inflammation in the initiation and progression of the disease remains unclear, there is plenty of evidence to indicate that IPF is associated with inflammation and changes in the innate and adaptive immune response [40]. Studies have shown that expression of inflammatory chemokine (C-C motif) ligands (CCL) such as CCL2 [41], CCL11 [42], and CCL8 [43] are elevated in fibrotic lungs. The secretion of the pro-inflammatory cytokine, IL-1β, has been linked to the progression and development of fibrosis by enhancing the expression of the inflammatory mediators interleukin 6 (IL-6) and tumor necrosis factor α (TNF-α), disrupting alveolar architecture and increasing pulmonary fibroblasts and collagen deposition [44]. Release of the pro-fibrotic cytokines TGF-β1 and PDGF can also be stimulated by IL-1β in BAL fluid [44]. IL-1β has also been shown to increase the infiltration of neutrophils and macrophages to the lungs [44,45] and elevate the expression of matrix metalloproteinases (MMPs) MMP-9/12 and chemokine (C-X-C motif) ligands (CXCL) CXCL1/2 [45].

Inflammatory cells found in the lungs of IPF patients have been shown to produce elevated levels of reactive oxygen species (ROS) which are thought to contribute to tissue damage in IPF [46,47]. Mitochondria-derived ROS can also drive pro-inflammatory cytokine production including IL-1β expression [48]. ROS production has been shown to be regulated by TGF-β1 and to further mediate downstream cellular events such as IL-6 expression [49] and activation of PAI-1 [50], an important regulator of ECM degradation. As such, in this environment where fibrotic and inflammatory mediators work together, an acute lung injury can readily escalate into a chronic fibrotic response; thus, controlling the acute inflammatory activity could prove beneficial by eliminating the downstream effects of a chronic progressive fibrotic state.

The adaptive immune response has also been linked to the development of IPF. Interestingly, the pro-inflammatory cytokine interleukin 17A (IL-17A) which is expressed by CD4^+^ T-helper (T_H_-17) cells, has been linked with enhanced neutrophil recruitment, also known as neutrophilia, as well as TGF-dependent and IL-1β-driven fibrosis [51]. Notably, a study by Kinder lncRNAs demonstrated that elevated levels of neutrophils in BAL fluid was shown to be a prognostic predictor of early mortality in IPF patients [52]. Additionally, T_H_-1 effector T cells are thought to exert anti-fibrotic activities through the production of IFN-γ, which has been shown to attenuate fibrosis [53] by inhibiting TGF-β-induced phosphorylation of Smad3 [54]. T_H_-2 effector T cells are thought to promote fibrosis via the production of pro-inflammatory cytokines such as interleukin 13 (IL-13) which stimulates collagen deposition in fibroblasts [55].

In this environment, activated platelets, damaged epithelial cells, and recruited inflammatory cells release more pro-fibrotic growth factors and such TGF-β1 drive the fibrotic response. TGF-β1 is considered the driving force of fibrosis as it has multiple properties including activation of fibroblasts, epithelial cell differentiation via epithelial-mesenchymal transition (EMT), and stimulation of the expression of pro-inflammatory cytokines such as IL-1β to further enhance the fibrotic response [56].

## 6. The Fibrotic Response and ECM Remodeling Phase

The aberrant wound healing response in IPF involves several underling mechanisms which are thought to drive the disease. However, the most well-established and studied concept is the role of TGF-β1 in the development of fibrosis and its effect on fibroblasts and the ECM which play a major role in the fibrotic cascade [57]. TGF-β1 activates a complex network of intracellular pathways and exhibits several properties that are thought to promote fibrosis, such as EMT [58], apoptosis [59], as well as recruitment and proliferation of fibroblasts via PDGF expression which can also release more TGF-β1 [60]. TGF-β1 is secreted by most cells, however AECs are the main source of TGF-β1 during the initial stages of wound repair. TGF-β1 can also regulate the expression of several pro-inflammatory and pro-fibrotic mediators and work with them synergistically to further enhance the fibrotic response [61].

Most importantly, in this fibrotic environment, both fibroblasts and myofibroblasts secrete increased amounts of ECM components to synthesize and maintain the ECM. The ECM is a complex and versatile network of cross-linked and fibrous proteins that form a protective structure under healthy conditions; however, the excessive deposition of the matrix in IPF is thought to exert powerful effects on cell functions via ECM-cell interactions [62]. As such, secreted TGF-β1 has been shown to be a potent inducer of ECM production [63], whereas the mechanical stress of ECM and contractile myofibroblasts further stimulate the activation of TGF-β1 [64]. Fibroblasts are also capable of matrix degradation by producing MMPs and the tissue inhibitors of metalloproteinases (TIMPs) (Figure 2). However, the highly complex nature of ECM biology is reflected by the diverse roles of MMPs and TIMPs in fibrosis where they demonstrate both pro- and anti-fibrotic activities during the tissue remodeling phase [65]. Interestingly, the expression of several ECM-degrading enzymes has been shown to be elevated in the IPF lungs, however the excessive deposition of ECM components results in the accumulation and the development of a stiff matrix and ultimately lung fibrosis [66,67].

## 7. Genetic Studies in IPF

IPF is a heterogeneous disease that is characterized by complex genetic and environmental interactions that contribute to the development of the disease; and therefore it is now thought that genetically susceptible individuals exposed to environmental stressor stimuli have an increased risk of developing the disease. Both rare and common genetic variants have been associated with sporadic and familial pulmonary fibrosis [68].

Genetic studies have linked IPF in adults to rare genetic variants in surfactant protein-related genes. Mutations have been found in the genes such as surfactant protein C (SFTPC) and A2 (SFTPA2) [69] which can result in alveolar epithelial cell injury following a disruption in their synthesis. Specifically, it appears that the SFTPC mutation causes defects in protein folding within the endoplasmic reticulum (ER) of type II AECs, which has been associated with IPF progression [70]. Similarly, the SFTPA2 mutation has been shown to enhance ER stress and has been associated with the development of pulmonary fibrosis [71,72].

Rare variants in several genes regulating telomere biology have also been identified in IPF patients, particularly mutations affecting telomerase and telomerase-associated proteins such as TERT [69,73] and TERC [74]. Telomeres are found at the end of chromosomes and protect them from DNA damage during the replication process. TERT and TERC encode telomerase genes which restore telomere length and mutations lead to increased telomere shortening. Interestingly, IPF has been associated with telomere shortening [75], although the exact mechanisms that links this to the fibrotic response are still unclear.

The single-nucleotide polymorphism rs35705950, located in the putative promoter region of the MUC5B gene, was also found to play a role in predisposing patients to familial and sporadic forms of IPF by causing increased MUC5B expression [76]. The MUC5B gene encodes for mucin 5B, a gel-forming protein, expressed by epithelial cells. However, upregulated MUC5B expression is also observed in IPF patients in the absence of rs35705950 indicating alternative mechanisms that can increase MUC5B expression, which are currently being investigated [77].

## 8. Gene Expression Studies in IPF

Transcriptional changes in the lungs of IPF patients have also contributed to our understanding of the disease and its underlying molecular mechanisms. In particular, microarray and sequencing-based approaches have identified an association between IPF and ECM formation, smooth muscle markers, growth factors, chemokines, and immunoglobulins [78].

Several gene expression profiling studies have demonstrated widespread changes in the profile of mRNA (messenger RNA) expression in lung biopsies. A study by Nance et al., identified 873 differentially expressed genes in IPF lung biopsies as compared with controls using RNA sequencing (RNA-seq). Interestingly, 675 of these genes displayed alternative splicing events including those coding for periostin (POSTN) and collagen (COL6A3) [79]. Microarrays showed differential expression of 2940 genes in IPF lung tissue as compared with controls, including genes encoding for collagens, proteinases, cytokines, and growth factors [80]. The gene expression profile of control and IPF lung biopsies was also assessed using microarrays by Bridges et al [81]. The Twist1 gene was the most consistently upregulated in IPF lungs and was found to have a protective role against apoptosis. Gene expression was also assessed in the lungs of IPF patients to identify mechanisms of acute exacerbations. This study identified 579 differentially expressed genes, including CCNA2 and α-defensins which were amongst the most upregulated genes [82]. In another study, comparison of lung biopsies from relatively stable and progressive IPF patients demonstrated differential expression of 243 transcripts including CCL2 and SFTPA1 [83].

As well as biopsies, gene expression in isolated human lung fibroblasts has also been employed to examine and identify novel IPF-related genes and pathways. A recent microarray study identified several IPF-associated genes upregulated in both lung control and IPF fibroblasts in response to TGF-β1 [84]. Another study by Lee et al., identified CCL8 expression to be elevated in IPF lung fibroblasts using microarrays [43]. Another report by Plantier et al., used publicly available microarray data to analyze and compare the expression of genes in cultured control and IPF fibroblasts [85]. Notably, two of the most significantly expressed factors were the connective tissue growth factor (CTGF) and serum response factor (SRF) which were shown to be overexpressed in IPF fibroblasts. Gene expression of lung fibroblasts was also assessed by microarrays following 4 h treatment with TGF-β1 [86]. The expression of 129 transcripts was shown to be driven by TGF-β1 stimulation including SMAD specific E3 ubiquitin protein ligase 2 (SMURF2), bone morphogenetic protein 4, and angiotensin II receptor type 1 (AGTR1).

In an attempt to identify circulating biomarkers of IPF, transcriptome analysis has also been undertaken on blood serum and plasma. A study by Yang et al. [87] used microarrays to evaluate circulating genes in IPF patients based on disease severity. The study identified 1428 differentially expressed transcripts in the peripheral blood of mild IPF and 2790 differentially expressed transcripts in severe IPF as compared with control patients. The genes encoding for MMP9 and Il-1R2 were found to be upregulated in both mild and severe IPF patients. Elevated levels of systemic MMP3 and CXCL13 were also identified in the blood of IPF patients using microarrays [80].

## 9. Current Pharmacological Strategies for the Treatment of IPF

There are currently two drugs that have been approved for the treatment of IPF, nintedanib and pirfenidone, both of which slow the decline of lung function associated with the disease.

Nintedanib, also known as BIBF1120, was originally developed as an anticancer agent and is thought to be a nonspecific tyrosine kinase inhibitor. A study by (Hilberg et al) [88] showed intracellular inhibition of the receptors for vascular endothelial growth factor (VEGFR), fibroblast growth factor (FGFR), and platelet-derived growth factor (PDGFR) by nintedanib. Evidence that nintedanib has anti-fibrotic activities came from reports showing attenuated fibrosis in the bleomycin-induced model of lung fibrosis and inhibition of TGF-β-induced fibroblast to myofibroblast differentiation in vitro [89,90]. It was also shown to inhibit collagen deposition induced by TGF-β in human lung fibroblasts in vitro [91]. Nintedanib could also have anti-inflammatory activity following the observation that it reduced Il-1β production and lymphocyte counts in bronchoalveolar lavage fluid (BALF) obtained from human fibrotic lungs [90]. In 2014, nintedanib was approved by the US Food and Drug Administration (FDA) for oral use in IPF patients after the completion of the INPULSIS-1 and IMPULSIS-2 trials [92]. The IMPULSIS trials were two 52-week parallel phase III randomized double-blind trials to evaluate its safety and efficacy in a total of 1066 patients. Overall, nintedanib was shown to reduce the decline of FVC and was considered safe, with no significant increase in mortality rates as compared with the placebo group. Mild to moderate diarrhea was reported to be the most frequent adverse effect experienced by patients along with other events such as nausea and vomiting.

Pirfenidone is thought to be an antioxidant and anti-inflammatory compound which was first shown to have anti-fibrotic activities in the bleomycin model of lung fibrosis [93]. It was suggested to exert an anti-fibrotic effect by targeting and inhibiting TGF-β expression [94]. Pirfenidone was administered to terminally ill IPF patients for the first time in a clinical setting as part of a small open-label phase II study by Raghu et al., which showed promising results with satisfactory tolerability and relatively minor adverse events [95]. Pirfenidone has been shown to reduce levels of monocyte chemoattractant protein (MCP) -1, TGF-β, IFN-γ, FGF, IL-6, IL-1β, and other cytokines in the bleomycin-fibrosis model [96]. Human lung fibroblast proliferation and TGF-β-induced differentiation is also inhibited by pirfenidone in vitro. Phosphorylation of Smad3, p38, and Akt, which are key molecules in the TGF pathway, were also decreased by pirfenidone [97], although its exact mechanism of action is still unclear. Pirfenidone was approved as the first anti-fibrotic therapy for IPF following the demonstration of improved mortality and reduced disease progression during the CAPACITY [98] and ASCEND trials [99]. The CAPACITY trials (studies 004 and 006) evaluated the oral administration of pirfenidone for at least 72 weeks. Paradoxically, although one trial (004) showed an overall reduction in the decline of FVC, this was not observed in the other trial (006) [98]. This led to the ASCEND trial which evaluated the administration of pirfenidone in 555 patients over 52 weeks. In this trial, pirfenidone was shown to slow the decline in lung function, improve exercise tolerance, and to be generally safe and well tolerated with an acceptable adverse event profile [99].

## 10. Non-Coding RNAs

Initial data from the Human Genome Project indicated that there were approximately 30,000 to 40,000 protein coding genes in the human genome [100]. However, this estimate was significantly reduced to 20,000 to 25,000 in the final draft published three years later [101]. In terms of the total length of the human genome, this meant that just 1% to 2% codes for protein coding genes. Although the function of the remaining DNA is currently an area of investigation, we know that much of this is transcribed into RNA which is not translated, and therefore is classified as “non-coding RNA” (ncRNA) [102,103]. Despite being initially considered as “junk”, ncRNAs are now known to have multiple biological functions and play a significant role in health and disease [104,105].

The term ncRNA includes all RNA molecules that do not encode for proteins and a very big proportion of these are the well-known “housekeeping” RNAs including transfer RNAs and ribosomal RNAs, which play a critical role in protein biosynthesis. In addition, there are the small nucleolar RNA (snoRNA), which regulate the transcriptional modification of other RNA species and the small nuclear RNAs (snRNA) that form an important component of the spliceosome complex which removes introns from mRNA during transcription. In total, these “housekeeping RNAs” represent around 85% to 90% of the total RNA whilst the mRNAs account for 5% to 8%. The remaining ncRNAs are speculated to have regulatory roles in various cellular functions and are divided in two classes, i.e., short (<200 nucleotides) which are exemplified by the microRNA (miRNA) and long (>200 nucleotides) non-coding RNAs (lncRNAs) [102]. MicroRNAs are thought to predominantly regulate gene expression at the translational level and lncRNAs at the transcriptional level (Figure 3).

## 11. Long Non-Coding RNAs

Despite the rapid increase in data relating to lncRNAs, little is known regarding their exact functions, mechanism of action, or even how many different types of lncRNAs exist. Therefore, due to the lack of mechanistic studies, lncRNAs are mainly grouped according to their proximity to protein coding genes. The most significant groups are lincRNAs (long intergenic non-coding RNAs), antisense, intronic, enhancers, and pseudogenes [106]. Although these transcripts are generally poorly evolutionary conserved [107], it is now evident that they hold a critical role in multiple biological pathways, including the modulation of several developmental processes and pathophysiological states [104,108], by regulating transcription of protein coding genes via a variety of mechanisms.

## 12. Chromatin Modifications

Long non-coding RNAs have been implicated in the regulation of epigenetic changes through recruiting and guiding chromatin remodeling complexes to specific genomic loci both in cis and in trans to regulate transcription (Figure 4). The chromatin remodeling complexes can exert their effects by repressing or activating the expression of protein coding genes via the recruitment of chromatin-modifying factors such as the Polycomb repressive complexes (PRC1 and PRC2) and histone methyltransferases.

For example, one of the first lncRNAs to be characterized was the mammalian cis-acting XIST which mediates the silencing of one X chromosome in females during development. XIST induces the formation of repressive chromatin and the recruitment of PRC proteins (PRC2) to completely inactivate one of the two X chromosomes by dosage compensation during the early embryonic development of females. Interestingly, other lncRNAs have been shown to interact with and regulate the expression of XIST, with the most prominent being its own natural antisense lncRNA (TSIX) [109,110,111]. Another well-studied lncRNA is HOTTIP (HOXA distal transcript antisense RNA) which was found to promote the expression of the HOXA gene. HOTTIP directly interacts with adaptor protein WDR5 and the mixed-lineage leukaemia protein 1 (MLL1) histone lysine methyltransferase complex (also known as histone lysine N-methyltransferase 2A) to recruit them to the HOXA locus through chromatin looping, causing H3K4 trimethylation and inducing the transcription of HOXA [112].

Long non-coding RNAs can also migrate from their site of transcription and regulate the expression of genes in trans either distally located on the same chromosomes or on different chromosomes. Such a lncRNA is the well-studied antisense intergenic lncRNA HOTAIR which is transcribed from the HOXC locus [113]. HOTAIR is thought to silence the transcription of the distant HOXD gene by acting as a scaffold for the recruitment of the repressive chromatin PRC2 complex leading to H3K27 trimethylation and H3K4 demethylation [114,115].

Long non-coding RNAs have also been associated with the regulation of the monoallelic expression (only one of the two copies of a gene is expressed) of genes according to their parents of origin, also known as genomic imprinting. Imprinted control regions may be associated with the expression of lncRNAs, such as the paternally expressed Airn lncRNA, also known as IGF2R-AS1, which was shown to silence the maternally expressed genes Igf2r/Slc22a2/Slc22a3 [116]. The imprinted lncRNA clusters are thought to silence the expression of neighboring genes in cis by recruiting chromatin-modifying complexes and maintaining repressive DNA methylation at adjacent loci [117].

## 13. Transcriptional Regulation

The process of transcription and the associated RNA processing and organization of the nuclear architecture, can also be regulated by lncRNA transcripts. The lncRNAs have been shown to regulate transcription factors and the RNAPII transcription machinery, leading to an increase or suppression of transcription, as well as mRNA processing mechanisms including splicing, capping, and editing [118]. For example, the lncRNA MALAT1 is found in nuclear speckles and is thought to regulate alternative splicing of mRNAs [119]. MALAT1 can act as a scaffold by guiding serine and arginine (SR) splicing factors to sites of transcription where splicing takes place [105]. Another example is the natural antisense transcript ZEB2 (NAT) which was shown to regulate the expression of the Zeb2 gene via splicing of an internal ribosome entry site (IRES) located within an intron in the 5-untranslated region (UTR) of the Zeb2 gene [120].

## 14. Post-Transcriptional Regulation

Several lncRNAs are transported from the nucleus to the cytoplasm where they are thought to be involved in regulating mRNA stability and translation. As an example, the ubiquitin carboxy-terminal hydrolase L1 antisense RNA 1 (Uchl1-as1) exhibits positive regulation of translation of the Uchl1 protein through the embedded inverted SINEB2 element, although its exact mechanism of action is unclear [121,122]. In contrast, the lincRNA-p21 (also known as tumor protein p53 pathway core pressor 1) was shown to inhibit the translation of target mRNAs by negatively regulating the translation of CTNNB1 (β-catenin) and JUNB (transcription factor jun-B) [123]. Long non-coding RNAs such as 1/2-sbsRNAs and gadd7 have also been shown to regulate mRNA stability through interactions with various proteins including Staufen1 (STAU1) and cyclin-dependent kinase 6 (Cdk6), respectively [124]. Additionally, lncRNAs such as the natural antisense BACE1-AS and TINCR were shown to enhance mRNA stability in vitro [125,126]. Interestingly, lncRNAs may also regulate mRNA expression by binding to specific miRNAs where they function as competing endogenous RNAs (ceRNAs) which have been shown to protect target mRNAs from repression by acting as ”miRNA sponges”, presenting another post-transcriptional regulatory role of lncRNAs [125,127,128].

## 15. Long Non-Coding RNAs and the Regulation of Lung Fibrosis

Although our knowledge of how miRNAs regulate disease, including IPF, is well established [129]; our knowledge regarding the role of lncRNAs in IPF is currently very limited. However, due to the complexity of this disease, more studies are now focused on investigating the role of lncRNAs in the development of IPF. All documented lncRNAs involved in the pathogenesis of pulmonary fibrosis are discussed below and listed in (Table 1).

## 16. Long Non-Coding RNAs AJ005396 and S69206

A microarray study by Cao et al. [130] using the bleomycin-induced lung fibrosis rat model was one of the first to identify differential expression levels of multiple lncRNAs and mRNAs, most significantly lncRNAs AJ005396 and S69206. In situ hybridization confirmed the expression of these lncRNAs and located their expression in the cytoplasm of interstitial lung cells.

## 17. Long Non-Coding RNAs MRAK088388 and MRAK081523

A subsequent study using the same model focused on two differentially expressed lncRNAs, MRAK088388 and MRAK081523, reporting that these could regulate expression of protein coding genes, N4bp2 (Nedd4 binding protein 2) and Plxna4 (plexin-A4), by acting as ceRNAs (sponges) for miR-29b-3p and let-7i-5p. In situ hybridization also confirmed their expression in the cytoplasm of interstitial lung cells [131].

## 18. Long-Non-Coding RNAs CD99P1 and n341773

Huang et al. [132] identified 34 lncRNAs with potential miRNA binding sites suggesting possible lncRNA–miRNA interactions. Real-time polymerase chain reaction (RT-PCR) confirmed the expression of these lncRNAs in human IPF lung tissue and nine of them were dysregulated in IPF. Knockdown of lncRNAs CD99P1 (CD99 molecule pseudogene 1) and n341773 indicated that these could regulate lung fibroblast collagen and α-SMA expression, differentiation, and proliferation. 

## 19. Long Non-Coding RNAs uc.77 and 2700086A05Rik

More recently, sequencing and microarray analysis of mouse fibrotic lungs identified 513 upregulated and 204 downregulated lncRNAs, of which uc.77 and 2700086A05Rik were found to mediate changes in EMT when overexpressed in human epithelial cells [133].

## 20. Long Non-Coding RNA CHRF

Wu et al., has demonstrated that silica-induced pulmonary fibrosis can be inhibited by miR-489 in mice. In addition, miR-489 was shown to suppress fibroblast differentiation and inflammation by targeting Smad3 and MyD88, respectively. Interestingly, the upregulation of the lncRNA CHRF (cardiac hypertrophy-related factor) was found to reverse the inhibitory effects of miR-489 in mice, as well as in macrophage and fibroblast cell lines. This suggests that CHRF can play a role in the regulation of miR-489 and the activation of the inflammation and fibrotic signaling pathways [134].

## 21. Long Non-Coding RNA H19

The expression of lncRNA H19 has also been implicated in the development of pulmonary fibrosis in a study by Tang et al. [135] who found that lncRNA H19 interacted with miR-29b and exerted pro-fibrotic effects by regulating the expression of collagen and α-SMA in the bleomycin model of fibrosis. However, another study demonstrated upregulated expression of lncRNA H19 in the bleomycin mouse model, as well as in fibroblast cell lines, following TGF-β1 exposure [136], where lncRNA H19 was found to act as a ceRNA for miR-196a to regulate collagen expression.

## 22. Long Non-Coding RNA AP003419.16

Moreover, the elevated expression of lncRNA AP003419.16 and its adjacent protein-coding gene RPS6KB2 (ribosomal protein S6 kinase B-2) in the peripheral blood of IPF patients has been linked with the increased risk of developing age-associated IPF [137].

## 23. Lon Non-Coding RNA NONMMUT065582 (PFAR)

Interestingly, the expression of lncRNA NONMMUT065582, also known as pulmonary fibrosis-associated RNA (PFAR), was found to be elevated in the fibrotic lungs of mice as well as mouse fibroblasts [138]. The LncRNA, PFAR, was shown to promote the development of fibrosis by acting as ceRNA for miR-138 and regulating the expression of yes-associated protein 1 (YAP1).

## 24. Long Non-Coding RNAs IL7AS and MIR3142GHG

More recently, expression of IL7AS and MIR3142GHG was also shown to be significantly upregulated in Il-1β-activated human control and IPF lung fibroblasts. Knockdown of IL7AS demonstrated a negative regulatory activity of this lncRNA in IL-6 mRNA expression and protein release (Figure 5), whereas MIR3142HG was found to positively regulate expression and protein release of CCL2 and Il-8. Both lncRNAs were found to be regulated by the NF-κB pathway [139].

## 25. Long Non-Coding RNAs LINC01140 and LINC0090

LINC01140 and LINC0090 were also found to be upregulated in isolated IPF fibroblasts, and LINC01140 was also upregulated in IPF biopsies and was shown to be a negative regulator of the inflammatory response [84]. Knockdown studies have also demonstrated that both lncRNAs play a role in mediating proliferation with and without PDGF-stimulation in both control and IPF fibroblasts.

## 26. Future Perspectives

Although IPF research has made impressive progress in the last decade, it is now evident that understanding the underlying mechanisms of IPF is much more challenging than originally thought. IPF is still an almost invariably lethal disease and a deeper understanding of its pathogenesis is essential for the development of novel and more effective therapies. LncRNAs present a relatively new group of molecules that have been found to regulate gene expression; however, little is known regarding their role in IPF. As a consequence of their cell and tissue-specific expression, the identification of lncRNAs that drive cellular activities within the IPF lungs could present a great opportunity for the development of novel treatment strategies. Several oligonucleotide antisense therapeutics have already proven successful in treating human disease and are being administered to patients, while nucleic acid-targeting drugs have demonstrated great potential in targeting lncRNAs in cancer [140]. Therefore, better delineating the functions of lncRNAs in the fibrotic lungs could, therefore, render them as potential targets for pharmacological intervention for IPF.

## Figures and Tables

**Figure 1 ijms-21-00524-f001:**
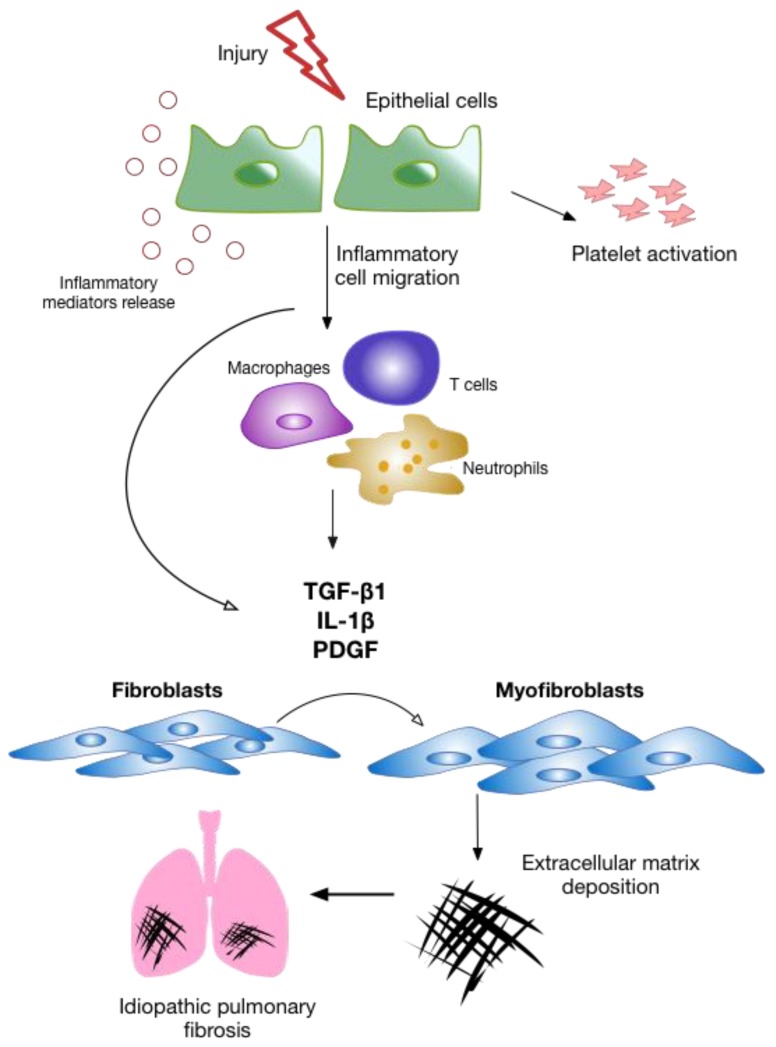
Overview of the most significant wound healing stages leading to the development of idiopathic pulmonary fibrosis (IPF). Epithelial cell injury leads to the secretion of inflammatory mediators and triggers platelet activation which results in enhanced vessel permeability for the recruitment of leukocytes. These inflammatory cells release pro-fibrotic cytokines such as TGF-β1 that mediate the activation and recruitment of fibroblasts as well as their differentiation into myofibroblasts and the subsequent release of extracellular matrix (ECM) components to promote wound healing. In IPF, an aberrant wound repair response leads to the irreversible formation of excessive scar tissue in the lungs.

**Figure 2 ijms-21-00524-f002:**
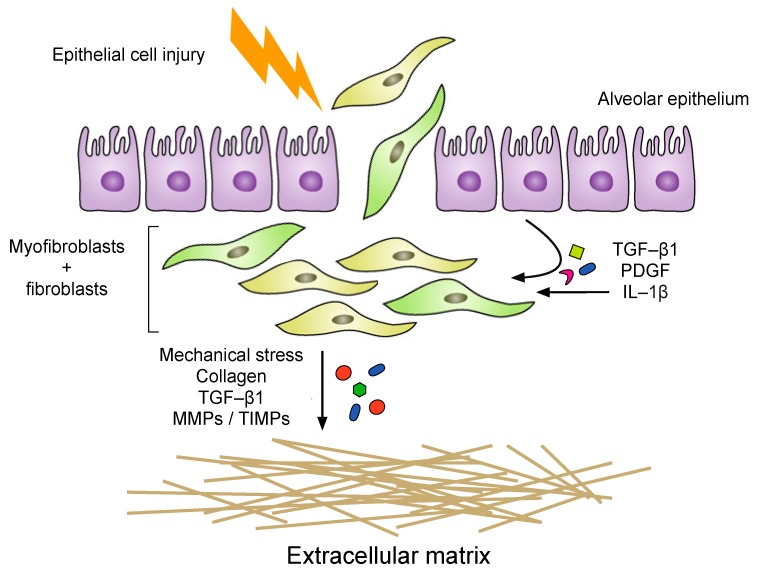
Pulmonary fibroblasts interact with the extracellular matrix to enhance the fibrotic response. During the fibrotic response, the alveolar epithelial cells undergo apoptosis due to injury which results in the infiltration of fibroblasts and myofibroblasts into the alveolar space. The dysregulation of fibroblasts is thought to be a critical player in the development of IPF where they excessively synthesize and release extracellular matrix components. The matrix is maintained by the presence of pro-fibrotic mediators released by the fibroblasts and other cells. The interactions of the fibroblasts and the ECM further enhance the fibrotic response in a positive-feedback loop.

**Figure 3 ijms-21-00524-f003:**
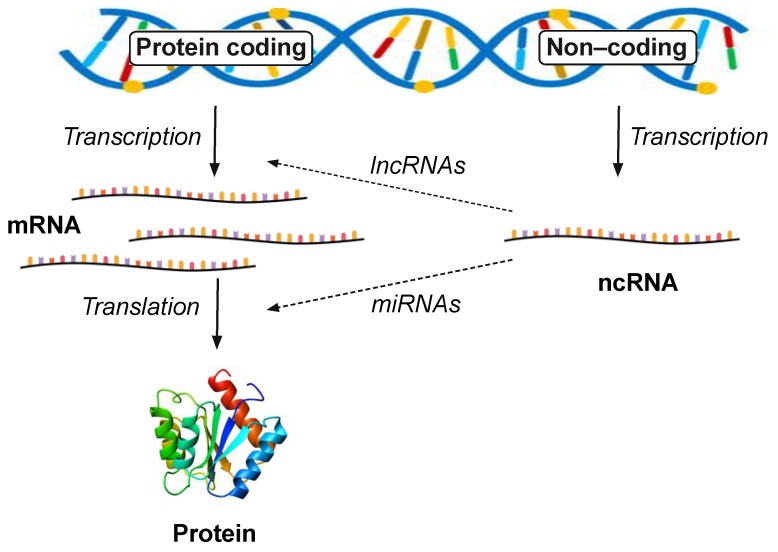
Long non-coding RNAs are novel modulators of transcription and lncRNA transcripts are thought to interfere with the expression of protein coding genes at the transcriptional level, whereas miRNAs are thought to silence the expression of genes at the translational level.

**Figure 4 ijms-21-00524-f004:**
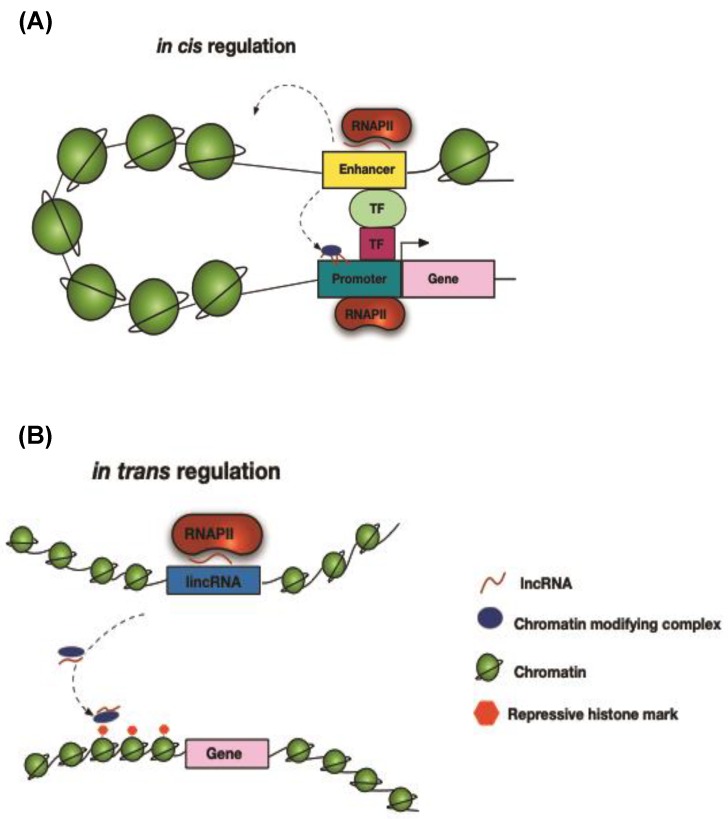
Long non-coding RNAs function in cis and in trans to regulate gene expression by interacting with chromatin remodeling factors and binding proteins in order to regulate the expression of (**A**) neighboring genes, in cis or (**B**) in trans, distally located genes that could be located on the same or a different chromosome. TF, transcription factor and RNAPII, RNA polymerase II.

**Figure 5 ijms-21-00524-f005:**
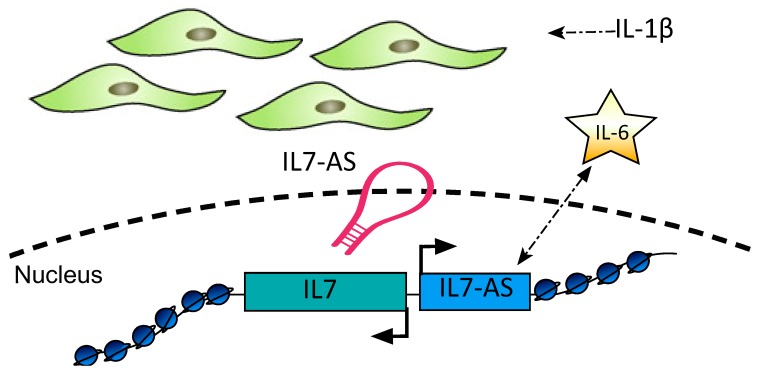
The antisense lncRNA, IL7AS, is upregulated upon IL-1β exposure in human fibroblasts. Knock-down of IL7AS was shown to elevate both Il-6 mRNA and protein levels, indicating a significant role of this lncRNA in mediating the inflammatory response in human lung fibroblasts.

**Table 1 ijms-21-00524-t001:** LncRNAs associated with pulmonary fibrosis.

LncRNA	Stimuli	Function	Research Model	Reference
**AJ005396, S69206**	Bleomycin	N/A	Sprague-Dawley (SD) rats	[130]
**MRAK088388, MRAK081523**	Bleomycin	Act as ceRNA for miR-29b-3p and let-7i-5p to regulate N4bp2 and Plxna4 expression	Sprague-Dawley (SD) rats	[131]
**CD99P1, n341773**	N/A	CD99P1 regulates proliferation and α-SMA expression n341773 regulates collagen expression	Human lung tissue, LL29 human lung fibroblasts	[132]
**uc.77, 2700086A05Rik**	Paraquat	Demonstrate regulation Zeb2 and Hoxa3 gene as well as induce expression of several EMT markers and cell morphology	BALB/c mouse model, A549 human lung epithelial cells and primary bronchial epithelial cells	[133]
**lncRNA CHRF**	Silica PMA TGF-β1	Negatively regulates miR-489 expression to promote silica-induced fibrosis	C57BL/6 mouse model, mouse macrophages (RAW 264.7) and fibroblasts (NIH3T3), human monocytes (THP-1) and fibroblasts (MRC-5)	[134]
**LncRNA H19**	Bleomycin	Regulates COL1A1 and ACTA2 expression and interacts with miR-29b	C57BL/6 mouse model, NIH3T3 mouse fibroblast cells	[135]
**LncRNA H19**	Bleomycin TGF-β1	Regulates COL1A1 expression by sponging miR-196a	C57BL/6 mouse model, human fibroblast (MRC-5) and kidney (HEK-293T) cell lines	[136]
**AP003419.16**	TGF-β1	Elevates the expression of RPS6KB2	Human venous blood, A549 human lung epithelial cells	[137]
**PFAR**	Bleomycin TGF-β1	Promotes fibrogenesis by modulating miR-138 expression	C57BL/6 mouse model, primary mouse fibroblasts	[138]
**Il7AS** **MIR3142HG**	IL-1β	Il7AS regulates Il-6 and MIR3142HG regulates IL-8 and CCL2 mRNA expression and protein release via a NF-κB-dependent pathway	Human primary lung fibroblasts	[139]
**LINC01140** **LINC00960**	IL-1β PDGF-AB	Both lncRNAs regulate proliferation and LINC01140 negatively regulates IL-6 release	IPF lung biopsies, human primary lung fibroblasts	[84]

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
