# Peer review of "Idiopathic Pulmonary Fibrosis: Pathogenesis and the Emerging Role of Long Non-Coding RNAs"

_ijms, 2020, doi:10.3390/ijms21020524_

Round 1
Reviewer 1 Report
Overall I enjoyed reading this review and the information presented was well described and accurate. My biggest concern is with the title of the manuscript. Little is known about long non coding RNAs in IPF and therefore there is not a lot of information available in the literature to discuss. However I feel that the section on LNCRNA was a relatively minor section as the manuscript was more of a general review on the pathogenesis of IPF with some focus on non coding RNAs. I think the title should better reflect this.
Two minor points.
1. Some minor grammatical mistakes and spacing issues in the text.
2. Wound repair should replace ‘fibrogenesis' on line 49.
Author Response
The authors provide a very nice manuscript regarding IPF pathogenesis. In my assessment, the manuscript has two overall components. First is a broad overview of clinical features and molecular and cellular mechanisms involved in IPF pathogenesis, and includes discussion of the role of anti-fibrotic therapy in patients with IPF. The authors do a very nice job with these discussions. A clear discussion of the conflicting data regarding the role of inflammation in the pathogenesis of IPF is also provided, in particular discussing that mechanisms of inflammation are universally present, but traditional anti-inflammatory therapies have been uniformly poorly effective. The second portion of the manuscript focuses on the potential role of non-coding RNAs in the pathogenesis of IPF, in particular the long non-coding RNAs, and is likewise well written and referenced. The figures are nicely presented, and provide a nice overview to supplement the text. Table 1 provides a very nice overview of the long non-coding RNAs which have been associated with pulmonary fibrosis in vivo and/or in vitro, and is organized and well-referenced. The manuscript is well-written and very clear.
Minor concern:
The authors make the following statement near the very end of the manuscript: “several oligonucleotide antisense therapeutics have already proven successful and are being administered to patients.” Is this statement in regards to patients with pulmonary fibrosis, or patients with cancer? The reference provided seems to refer to patients with cancer. If these types of therapies have been administered to patients with pulmonary fibrosis, it would be helpful if the authors could provide references in this regard. If not, perhaps just clarifying this statement better to reflect either patients with cancer or patients with pulmonary fibrosis.
The statement has now been clarified.
Reviewer 2 Report
1/4/2020
International Journal of Molecular Sciences
Manuscript ID: ijms-684795
Title: The role of long non-coding RNAs in the pathogenesis of idiopathic pulmonary fibrosis
Authors: Marina R Hadjicharalambous , Mark A Lindsay
The authors provide a very nice manuscript regarding IPF pathogenesis. In my assessment, the manuscript has two overall components. First is a broad overview of clinical features and molecular and cellular mechanisms involved in IPF pathogenesis, and includes discussion of the role of anti-fibrotic therapy in patients with IPF. The authors do a very nice job with these discussions. A clear discussion of the conflicting data regarding the role of inflammation in the pathogenesis of IPF is also provided, in particular discussing that mechanisms of inflammation are universally present, but traditional anti-inflammatory therapies have been uniformly poorly effective. The second portion of the manuscript focuses on the potential role of non-coding RNAs in the pathogenesis of IPF, in particular the long non-coding RNAs, and is likewise well written and referenced. The figures are nicely presented, and provide a nice overview to supplement the text. Table 1 provides a very nice overview of the long non-coding RNAs which have been associated with pulmonary fibrosis in vivo and/or in vitro, and is organized and well-referenced. The manuscript is well-written and very clear.
Minor concern:
The authors make the following statement near the very end of the manuscript: “several oligonucleotide antisense therapeutics have already proven successful and are being administered to patients.” Is this statement in regards to patients with pulmonary fibrosis, or patients with cancer? The reference provided seems to refer to patients with cancer. If these types of therapies have been administered to patients with pulmonary fibrosis, it would be helpful if the authors could provide references in this regard. If not, perhaps just clarifying this statement better to reflect either patients with cancer or patients with pulmonary fibrosis.
Author Response

(The authors gave the same response as above.)
